# Protective Effect and Mechanism of Soybean Insoluble Dietary Fiber on the Color Stability of Malvidin-3-*O*-glucoside

**DOI:** 10.3390/foods11101474

**Published:** 2022-05-19

**Authors:** Yang He, Dongxia Chen, Yuheng Liu, Xiaozhen Sun, Wenrui Guo, Lingyu An, Zhenming Shi, Liankui Wen, Zhitong Wang, Hansong Yu

**Affiliations:** 1College of Food Science and Engineering, Jilin Agricultural University, Changchun 130118, China; heyang200704@jlau.edu.cn (Y.H.); chendongxia1218@163.com (D.C.); lyhh5689@163.com (Y.L.); sunxiaozhen825@163.com (X.S.); gwr13694434337@163.com (W.G.); aly18104373339@163.com (L.A.); szm17767897978@163.com (Z.S.); wenliankui@jlau.edu.cn (L.W.); 2Division of Soybean Processing, Soybean Research & Development Center, Chinese Agriculture Research System, Changchun 130118, China

**Keywords:** anthocyanins, insoluble dietary fiber, stable system, inclusion complex, hydrogen bonding

## Abstract

Anthocyanins have great health benefits, especially malvidin. *Vitis amurensis Rupr* are rich in malvidin, and malvidin-3-*O*-glucoside (Mv3G) monomer is the most abundant. However, natural anthocyanins are unstable, which limits their wide application in the food field. Soybean insoluble dietary fiber (SIDF) has high stability, and it can be used as an inert substrate to construct a stable system, which may improve the stability of anthocyanins. The optimal condition to construct a stable system of SIDF and Mv3G at pH 3.0 was determined by an orthogonal experiment. The results indicated that SIDF effectively improved the stability of Mv3G under different pH values (1.0~7.0), high temperature (100 °C for 100 min), and sunlight (20 ± 2 °C for 30 d) conditions. The absorption peak intensity of the UV–VIS spectrum of SIDF-Mv3G was enhanced, which indicated that there was interaction between SIDF and Mv3G. Fourier transform infrared spectroscopy analyses revealed that the -OH stretching vibration peak of SIDF-Mv3G was changed, which indicated that the interaction between SIDF and Mv3G was due to hydrogen bonding. X-ray diffraction analysis showed that the crystalline morphology of SIDF was opened, which was combined with Mv3G, and SIDF made Mv3G change to a more stable state. Scanning electron microscope analysis showed that SIDF and Mv3G were closely combined to form an inclusion complex. Overall, this study provides valuable information for enhancing the color stability of anthocyanins, which will further expand the application of anthocyanins in the food field.

## 1. Introduction

Anthocyanins are gorgeous in color and have many human health benefits, including antioxidant, antiobesity, and antidiabetes [1,2,3]. In recent years, people’s awareness of the relationship between diet and health has gradually increased, resulting in an increase in consumer demand for foods containing natural anthocyanins. Therefore, anthocyanins are widely used in the fields of food, natural pigments, cosmetics, and medicine [4]. The distribution of the six most common anthocyanins in nature is cyanidin, delphinidin, pelargonidin, peonidin, petunidin, and malvidin, accounting for 50%, 12%, 12%, 12%, and 7%, 7% of total anthocyanins, respectively [5]. Malvidin has high biological activity, but the overall distribution is less in nature [5]. It is worth noting that the high-yielding *Vitis amurensis Rupr* in the Changbai Mountains of China are rich in anthocyanins (the average content is above 150 mg/100 g·FW, and individual varieties are as high as 400 mg/100 g·FW) [6,7]. Malvidin accounts for 55~65% of total anthocyanins from *Vitis amurensis Rupr*, among which malvidin-3-O-glucoside (Mv3G) is the most abundant monomer [8]. There are many phenolic hydroxyl groups on the carbon skeleton of anthocyanins, so pH, light, heat, oxygen, metal ions, and other environmental factors can affect the color stability of anthocyanins [6,7]. The instability of anthocyanins greatly affects their application in food. Therefore, enhancing the color stability of anthocyanins is an important problem that needs to be solved urgently.

During food processing, polyphenols can interact with starch, protein, and cellulose to form a complex, which can improve the stability of polyphenols [9,10,11,12]. In particular, Quan et al. [13] found that adding soybean protein to purple-fleshed sweet potato anthocyanins could effectively improve the color stability at 100 °C. Sun et al. [14] reported that linear dextrin could be used as a food-grade carrier of curcumin. UV–VIS spectrophotometry (UV–VIS) and Fourier transform infrared spectroscopy (FT-IR) analyses showed that linear dextrin and curcumin formed an inclusion complex (a mixture in which the molecules of one component are contained in the crystal lattice of another component) by hydrogen bonding, which improved the stability of curcumin. Li et al. [15] also reported the interaction between soluble dietary fiber and polyphenols through conjugation and hydrogen bonding improved the stability of the polyphenols–soluble dietary fiber complex. Zhang et al. [16] microencapsulated ethylcellulose and polyphenols to diminish the instability of polyphenols as water-soluble compounds, particularly under harsh processing and storage conditions. Thus, the protective effect of macromolecules on polyphenols contributes to their application in more complex food systems.

Okara is the main byproduct during the processing of traditional soybean products such as tofu, soybean milk, and soybean oil. Okara is abundant in nutrients, especially soybean insoluble dietary fiber (SIDF), which accounts for 45~55% of the dry matter of okara [17,18]. SIDF has high stability and is not easy to react with other food ingredients, so it can be used as an inert substrate to build a stable system [19]. SIDF has a loose structure and rough surface, which can promote the interaction with polyphenols by adsorption or embedding [20]. Zhao et al. [21] reported that the hydration of insoluble dietary fiber (IDF) improved as its particle size decreased because of the greater surface area, increased number of polar groups, and exposure to other water-binding sites of IDF to the surrounding water. IDF has a high denaturation temperature, so the thermal characteristics are stable [22]. There is little research on the interaction between IDF and anthocyanins. It is concluded that SIDF has the potential to promote the stability of anthocyanins. In order to further explore the interaction between SIDF and Mv3G, this study explained the stabilization mechanism of SIDF and Mv3G, which will be more conducive to anthocyanins as stable pigments or functional components added to food.

This study aimed to optimize the stable system of SDIF and Mv3G and explain the stabilization mechanism. In this paper, the color stabilities of the stable system of SIDF-Mv3G (hereinafter referred to as SIDF-Mv3G) at different pH values, thermal conditions, or sunlight conditions were investigated. Furthermore, the interaction mechanism of SIDF and Mv3G was investigated through UV–VIS, FT-IR, X-ray diffraction (XRD), and scanning electron microscopy (SEM).

## 2. Materials and Methods

### 2.1. Material

The purity of Mv3G was ≥95%, which was extracted in our lab according to a previous study [8]. The purity of SIDF was ≥90%, which was extracted in our lab according to a previous study [23]. All chemicals used in this work were all analytical or HPLC grade.

### 2.2. Optimal Design of SIDF-Mv3G Stable System

The appropriate amount of SIDF and Mv3G was weighed and mixed in a certain proportion with PBS (citric acid–disodium hydrogen phosphate) at pH 3.0 to 100 mL (the final Mv3G concentration was 0.12 mg/100 mL). Under a certain temperature range, the stable system was prepared by stirring and emulsifying at 500 rpm/min.

According to the results of the pre-experiment, the SIDF–Mv3G ratio A (20:1, 30:1, 40:1; final sample concentration of Mv3G was 0.12 mg/mL), SIDF particle size B (200-, 300-, 400-mesh sieve), emulsification temperature C (20, 30, 40 °C), and emulsification time D (20, 30, 40 min) were optimized using an L_9_ (4^3^) orthogonal design. The stable effect was evaluated by measuring the absorbance at 521 nm and sedimentation eccentricity (S, %). Sedimentation eccentricity was calculated by Equation (1).
(1)$$S\;\left(\%\right)=\frac{W_{2}-W_{1}}{W_{3}-W_{1}}\times 100\%$$
where W_1_ is the weight of the centrifuge pipe, W_2_ is the weight of sediment added to the centrifuge pipe after centrifugation, and W_3_ is the weight of the centrifuge pipe plus sample.

The optimal conditions of the orthogonal experiment were selected to construct the stable system of SIDF-Mv3G. Mv3G solution and SIDF solution under the same conditions were used as the control. SIDF-Mv3G and Mv3G solutions were used for sunlight stability, thermal stability, and UV–VIS spectrum analyses. SIDF-Mv3G, SIDF, and Mv3G were lyophilized and used for pH stability, FT-IR, XRD, and SEM analyses.

### 2.3. Stability of SIDF-Mv3G2.3.1. pH Stability

#### 2.3.1. pH Stability

SIDF-Mv3G and Mv3G were dispersed in PBS solution with a pH of 1.0~7.0, respectively. Color changes of the SIDF-Mv3G and Mv3G solutions were measured using a Color Flex EZ colorimeter (HunterLab, American) with a white standard color plate as the background. Hunter color values (L*, a*, and b*) of the SIDF-Mv3G and Mv3G solutions were obtained, and the browning degree (ΔE) was calculated as follows:(2)$$\Delta E=\sqrt{{\left(L_{t}-L^{*}\right)}^{2}+{\left(a_{t}-a^{*}\right)}^{2}+{\left(b_{t}-b^{*}\right)}^{2}}$$
where L_t_ (51.85), a_t_ (−26.26), and b_t_ (14.32) are the L*, a*, and b* values of the standard white color plate.

#### 2.3.2. Thermal Stability

The samples were prepared in 5 mL glass tubes wrapped in tin foil, which were heated in a water bath at 100 °C for 100 min. Every 20 min, the samples were quickly cooled in ice. After 5 min, the absorbance of the upper supernatant was measured at 521 nm by UV–VIS spectrophotometry (The turbidity problem of an appropriate amount of SIDF is not serious, and the influ-ence is deducted by using the same amount of SIDF as the instrument benchmark in the experimental process). The retention rate (R, %) of anthocyanins was used as the evaluation index to evaluate the thermal stability.
(3)$$R\;\left(\%\right)=\left(\frac{A_{1}}{A}\right)\;\times 100$$
where A is the absorbance before sunlight or thermal stability treatment, and A_1_ is the absorbance after sunlight or thermal stability treatment.

#### 2.3.3. Sunlight Stability

The SIDF-Mv3G and Mv3G solutions were sealed in vials and irradiated with sunlight at room temperature (20 ± 2 °C; samples were placed on the indoor windowsill, and the daily insolation time was not less than 6 h). Samples were taken out after 0, 5, 10, 15, 20, 25, and 30 d. The absorbance of the upper supernatant was measured at 521 nm by UV–VIS spectrophotometry (The turbidity problem of an appropriate amount of SIDF is not serious, and the influence is deducted by using the same amount of SIDF as the instrument benchmark in the experimental process), and the retention rate of anthocyanins was calculated according to Equation (3)

### 2.4. Structural Characterization of SIDF-Mv3G

#### 2.4.1. UV–VIS Spectroscopy

The UV–VIS absorption spectra of the SIDF-Mv3G and Mv3G solutions were measured using a T6 UV-Vis spectrophotometer (Beijing Universal Instrument Co., Ltd., Beijing, China) in the wavelength range of 400~700 nm

#### 2.4.2. FT-IR Spectroscopy

The FT-IR spectra of the SIDF-Mv3G, SIDF and Mv3G were recorded using an IR Prestige-21 spectrometer (Shimadzu, Japan). The samples were blended with KBr at a ratio of 1:100 (*w*/*w*) and pressed into tablets for further measurement. The FT-IR spectra were measured in absorbance mode between 4000 and 500 cm^−1^.

#### 2.4.3. XRD

The XRD diffraction patterns of SIDF-Mv3G, SIDF, and Mv3G were collected using an X-ray diffractometer (D8 ADVANCE, Germany), equipped with copper Cu Kα radiation (λ = 0.154 nm) as the X-ray source (2θ range of 5~50°).

### 2.5. Statistical Analysis

All tests were repeated three times, and the results were expressed as the average value (±SD). SPSS was used for statistical analysis (ANOVA), and when *p* ≤ 0.05, the difference between the two groups was evaluated.

## 3. Results

### 3.1. Optimal Design of SIDF-Mv3G Stable System

The variation coefficient method, which assigns different weights to the indexes based on their degree of variation [24], is typically employed in the comprehensive evaluation of multiple indicators. The variation coefficient V*_i_* of the *i*th characteristic parameter value was calculated according to Equation (4). The weight value W*_i_* of the *i*th characteristic parameter was calculated according to Equation (5).
(4)$${\;V}_{i}=\frac{S_{i}}{X_{i}}$$
where V*_i_* is the variation coefficient, S*_i_* is the standard deviation, and X*_i_* is the arithmetic mean.
(5)$$W_{i}=\frac{V_{i}}{\sum_{i=1}^{n}V_{i}}$$
where W*_i_* is the weight, and $\overset{n}{\underset{i=1}{\sum }}V_{i}$ is the sum of *n* indexes.

According to Table 1, the weight of absorbance and sedimentation rate were 0.411 and 0.589, respectively, and the results showed that the sedimentation rate had a greater influence on the stable system of SIDF-Mv3G than absorbance. The optimal result of the test was A_2_B_1_C_2_D_3_ (1.246).

According to the *K* value, the order of the effect of the four factors on the stable system were C (emulsification temperature) > B (SIDF particle size) > A (SIDF–Mv3G ratio) > D (emulsification time), and the optimal condition was A_2_B_2_C_2_D_3_.

The variance analysis showed that the effects of the single factors of emulsification temperature were significant (*p* < 0.01) on the stable system, and the SIDF particle size and SIDF–Mv3G ratio exhibited were significant on the stable system (*p* < 0.05), whereas emulsification time was not significant on the stable system. In addition, micronization reduced the interfacial tension and could expose more polar groups, surface area, and water-binding sites to the surrounding water, which could be closely combined with Mv3G. For the above reasons, the optimal combination was adjusted to A_2_B_2_C_2_D_3_: SIDF–Mv3G ratio was 30:1 (*w/w*), SIDF particle size was a 300-mesh sieve, emulsification temperature was 30 °C, and emulsification time was 50 min. This result was confirmed by three parallel validations, and the actual score for the optimal condition was 1.387 ± 0.034. The relative standard deviation RSD = 0.024%, which showed that the results of the orthogonal design had good reproducibility.

### 3.2. Stability of SIDF-Mv3G

#### 3.2.1. pH Stability

Figure 1 shows the color changes of the SIDF-Mv3G and Mv3G solutions at different pH values (1.0~7.0). Specifically, both SIDF-Mv3G and Mv3G solutions changed from bright red to pink, and finally to purple. The color diversity of the solutions was caused by the structural changes of anthocyanins [6]. With the increase in pH value, the structures of the flavylium cation were destroyed and transformed into purple quinone base structures [25].

Color is one of the most important parameters of food and affects the application of anthocyanins. The subtle color changes of anthocyanins are difficult to judge by visual inspection. Therefore, the color of the SIDF-Mv3G and Mv3G solutions was described by the color measurement. As shown in Table 2, with the increase in pH value, the L*, a*, and b* values gradually decreased, which indicated that the anthocyanins had faded [26]. With the gradual increase in ΔE, the browning caused by pH was aggravated, and brown substances were produced due to the existence of oxygen. Compared with Mv3G, the L*, a*, and b* values of SIDF-Mv3G were higher, and ΔE was lower. The results showed that SIDF can effectively prevent Mv3G degradation and protect the color stability of Mv3G.

Generally, when the pH value of the anthocyanin solution is 3.0, the structure of anthocyanins can be transformed into a stable flavylium cation. However, when the pH value is lower than 3.0, some biological macromolecules cannot protect against anthocyanin degradation [10]. At the same time, a lower pH value could cause the taste quality of food to decrease. Koh et al. [27] reported that when the pH value was 2.6, the flavonoid cations of anthocyanins were more dominant in the electrostatic effect; however, the charge of pectin was minimal, which affected their interaction; when the pH value was 3.0, the interaction between them was enhanced. Most fruit juice drinks on the market have a pH value of about 3.0. If SIDF-Mv3G is applied to beverages, it will ensure the bright color of beverages, prolong the stability of anthocyanins, and have potential benefits to human health.

#### 3.2.2. Thermal Stability

The protective effect of SIDF on Mv3G at 100 °C is shown in Figure 2. Anthocyanins degraded rapidly at high temperature, and the structure of anthocyanins was opened and converted into chalcone, resulting in unpleasant brown substances [28]. After heating for 100 min, the retention rates of the SIDF-Mv3G and Mv3G solutions were 81.54% and 65.55%, respectively. The results showed that the retention rate of SIDF-Mv3G was significantly higher than the Mv3G solution, which indicated that SIDF-Mv3G had high thermal stability.

#### 3.2.3. Sunlight Stability

During food processing or shelf storage, food containing anthocyanins can fade and degrade under sunlight. Therefore, the retention rate of the SIDF-Mv3G and Mv3G solutions stored in natural light (20 ± 5 °C) for 30 d was measured to evaluate the sunlight stability of Mv3G. Figure 3 shows that the retention rates of the SIDF-Mv3G and Mv3G solutions in sunlight were 75.16% and 60.54%, respectively. The results showed that SIDF could prolong the storage time of Mv3G in sunlight.

### 3.3. Structural Characterization of SIDF-Mv3G

#### 3.3.1. UV–VIS Spectroscopy

Figure 4 shows the UV–VIS spectra of the SIDF-Mv3G and Mv3G solutions. The figure shows that the absorption peak intensity of SIDF-Mv3G was significantly higher than the Mv3G solution, which is due to the copigmentation between SIDF and Mv3G, leading to Mv3G moving to a stable flavylium cation structure [29]. In addition, the increasing absorbance might be due to the fact that polysaccharides provide nucleation sites for anthocyanins, and then Mv3G is adsorbed on SIDF [9].

#### 3.3.2. FT-IR Spectroscopy

If the FT-IR spectra shift obviously after adding other components to the matrix, it indicates that there is an obvious interaction between the components [30]. Figure 5 shows the FT-IR spectra of SIDF-Mv3G, SIDF, and Mv3G. Compared with SIDF and Mv3G, the -OH stretching vibration peak of SIDF-Mv3G moved from 3397 cm^−1^ and 3389 cm^−1^ to 3442 cm^−1^, indicating that the hydrogen bond between SIDF and Mv3G was increased [4]. The C-H stretching vibration of SIDF-Mv3G at 2806~2723 cm^−1^ was obviously weakened, indicating that the hydrophobic groups in SIDF were decreased [31]. The characteristic peak of the C-O angle deformation of phenols compounds remained at 1332 cm^−1^, which indicated that anthocyanins were combined into SIDF [4]. FT-IR spectra synthesis showed that SIDF was combined with Mv3G through hydrogen bonding as the main driving force.

#### 3.3.3. XRD

XRD is an important technique for structure detection [32]. Figure 6 shows the XRD patterns of SIDF-Mv3G, SIDF, and Mv3G. According to the XRD pattern of SIDF, the diffraction peaks at 14.67°, 15.11°, and 24.23° were typical cellulose diffraction peaks, which indicated that SIDF has a crystalline structure [33]. According to the XRD pattern of Mv3G, a broad peak appeared at 20.83°, which indicated that Mv3G has an amorphous structure [26]. Qin et al. [34] also found a similar amorphous structure in blueberry anthocyanins. In the XRD pattern of SIDF-Mv3G, the intensity of diffraction peaks around 14.67°, 15.11°, and 24.23° was weakened, and the absorption peak at 20.83° was weakened and widened, which indicated that the crystalline state of SIDF was opened and combined with Mv3G after mixing with SIDF [35,36]. Finally, SIDF made Mv3G change to a more stable state.

#### 3.3.4. SEM

Figure 7 shows the SEM results of SIDF-Mv3G, SIDF, and Mv3G. After the small particles of Mv3G were magnified, the surface was lamellar and irregular, and the specific surface area was large. SIDF had a loose and stratiform network structure, which might be due to the degradation of starch and protein caused by enzymatic hydrolysis in the extraction process [37]. The folding structure of SIDF was opened, and it may be better combined with anthocyanins. SIDF-Mv3G was an inclusion complex, which showed that the loose structure of SIDF was closely combined with the layered structure of Mv3G. SIDF wrapped Mv3G, thus protecting Mv3G from environmental factors such as different pH values, light, and high temperature.

Based on the above conclusion, there are two main reasons why SIDF improves the anythocyanin stability of Mv3G. The first reason why biological macromolecules improve the stability of polyphenols may be the interaction between biological macromolecules and polyphenols through hydrogen bonds, van der Waals forces, electrostatic action, and hydrophobic action. Gao et al. [38] reported that hydrogen bonding was the main driving force between (-)-epigallocatechin gallate and oat *β*-glucan, which was similar to our study. Guan et al. [39] reported that gum arabic and anthocyanins formed a complex through hydrophobic interaction, which improved the thermal stability of anthocyanins. This result was different from our research. This is because of the different structure of biological macromolecules, and the binding sites of macromolecules with anthocyanins are different, which result in different forces. Fernandes et al. [9] found in the beverage model that the complex between grape pectic polysaccharides and malvidin-3-*O*-*β*-D-glucoside was formed by electrostatic and hydrophobic interactions, which protected the chromophore of anthocyanins, thus possibly shielding the highly electrophilic C2-position of the flavylium cation. The C2 position was easily attacked by water due to chemical degradation, so protecting the C2 position can improve the stability of anthocyanins.

The second reason is that when SIDF is exposed to an aqueous solution, it can spontaneously combine with polyphenols in order to reduce SIDF contact with water. Meanwhile, SIDF has a network structure, which can adsorb and form an inclusion complex with anthocyanins. A study by Cai et al. [4] showed that the thermal stability of blueberry anthocyanins was improved by encapsulation with carboxymethyl starch and xanthan gum. Pradhan et al. [40] reported that the half-life of berberis lycium anthocyanins increased by 1.74 (60 °C), 1.28 (75 °C), and 1.44 (90 °C) times when anthocyanins were encapsulated by the hydrophobic cavity of *β*-cyclodextrin, thus improving the stability of anthocyanins.

## 4. Conclusions

The best stability conditions of SIDF and Mv3G were as follows: SIDF–Mv3G ratio was 30:1 (*w*/*w*), SIDF particle size was a 300-mesh sieve, emulsification temperature was 30 °C, and emulsification time was 50 min. When the pH value gradually increased from 1.0 to 7.0, the color of the SIDF-Mv3G and Mv3G solutions changed from rose-red to purple, which indicated that the structure of anthocyanins changed from flavylium cation to purple quinone base structures. ΔE increased with increasing pH, indicating that pH leads to the aggravation of anthocyanin browning, but SIDF could effectively prevent anthocyanins from browning. Meanwhile, the L*, a*, and b* values of SIDF-Mv3G were higher than the Mv3G solution, which indicated that SIDF could stabilize the color of anthocyanins. After the SIDF-Mv3G and Mv3G solutions were heated at 100 °C for 100 min, the retention rate of SIDF-Mv3G increased by 15.99% compared with the Mv3G solution. When the SDIF-Mv3G and Mv3G solutions were stored for 30 d at 20 ± 2 °C under sunlight, the retention rate of SIDF-Mv3G was 14.62% higher than the Mv3G solution, which indicated that SIDF could protect the color stability of anthocyanins under sunlight. The stable system of SIDF-Mv3G was mainly combined by hydrogen bonding to form an inclusion complex. This inclusion complex structure was relatively stable, which protected Mv3G from environmental factors, thus improving the stability of anthocyanins.

Considering the health benefits of anthocyanins and insoluble dietary fiber to the human body, stable SIDF-Mv3G can be applied to natural pigments and also be added to the food matrix as a functional factor. This study provides a new idea for the application of anthocyanins in the food industry.

## Figures and Tables

**Figure 1 foods-11-01474-f001:**
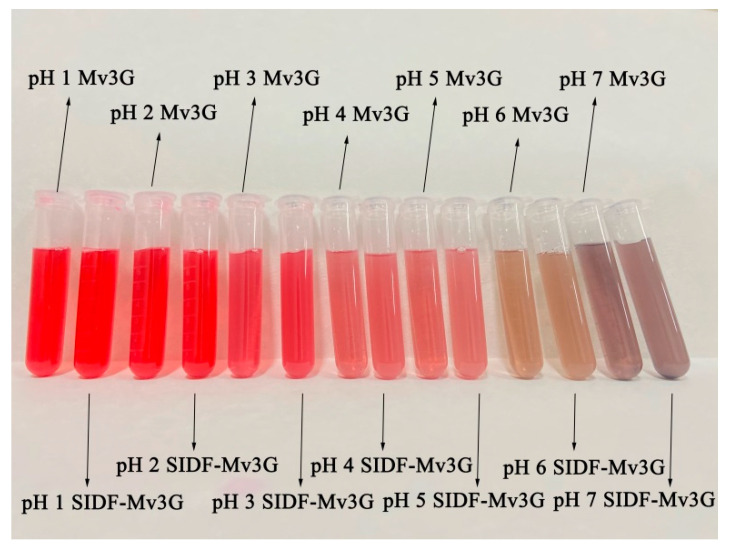
Color change of SIDF-Mv3G and Mv3G. SIDF-Mv3G: the stable system solution of soybean insoluble dietary fiber and malvidin-3-*O*-glucoside; Mv3G: malvidin-3-*O*-glucoside solution.

**Figure 2 foods-11-01474-f002:**
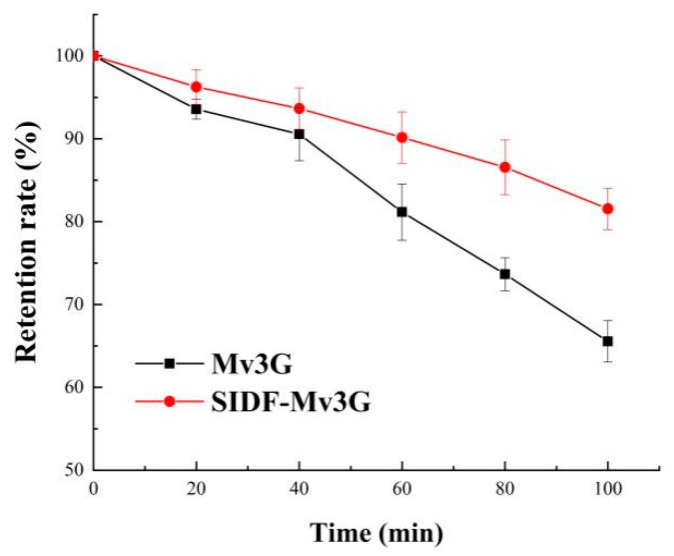
Retention rate of heat treatment at 100 °C for 100 min. SIDF-Mv3G: the stability system solution of soybean insoluble dietary fiber and malvidin-3-*O*-glucoside; Mv3G: malvidin-3-*O*-glucoside solution.

**Figure 3 foods-11-01474-f003:**
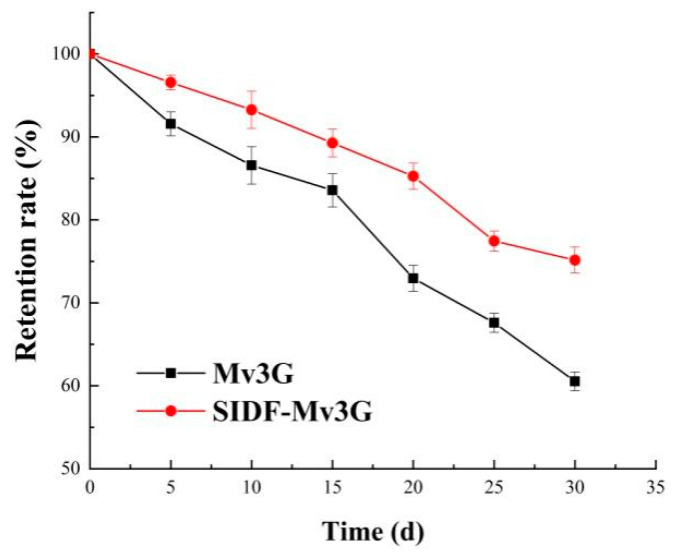
Retention rate of sunlight treatment at room temperature. SIDF-Mv3G: the stability system solution of soybean insoluble dietary fiber and malvidin-3-*O*-glucoside; Mv3G: malvidin-3-*O*-glucoside solution.

**Figure 4 foods-11-01474-f004:**
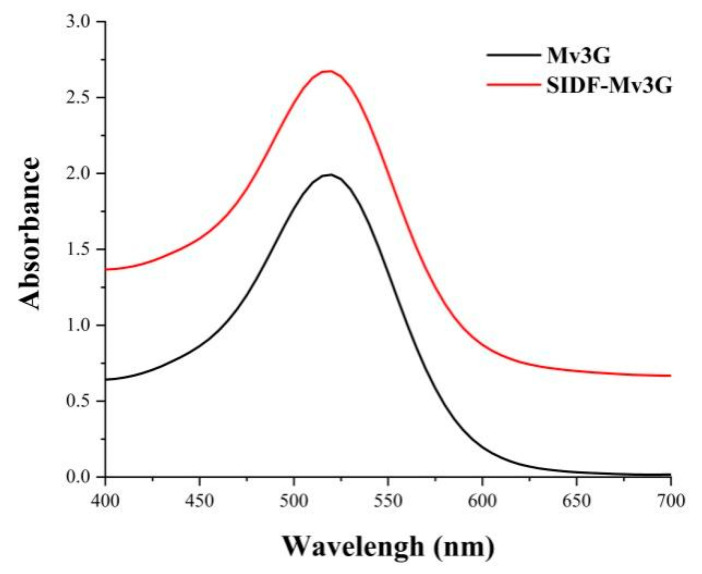
Ultraviolet–visible spectra. SIDF-Mv3G: the stability system solution of soybean insoluble dietary fiber and malvidin-3-*O*-glucoside; Mv3G: malvidin-3-*O*-glucoside solution.

**Figure 5 foods-11-01474-f005:**
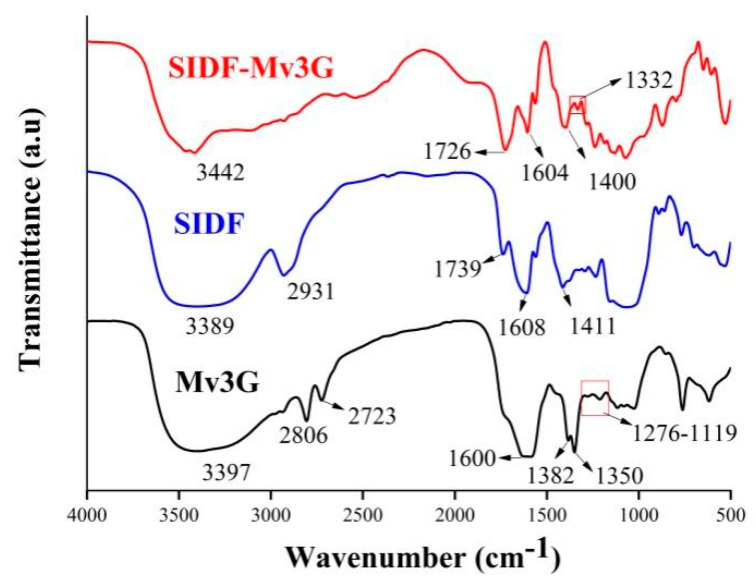
Fourier transform infrared spectra. SIDF-Mv3G: the stability system of soybean insoluble dietary fiber and malvidin-3-*O*-glucoside; Mv3G: malvidin-3-*O*-glucoside; SIDF: soybean insoluble dietary fiber.

**Figure 6 foods-11-01474-f006:**
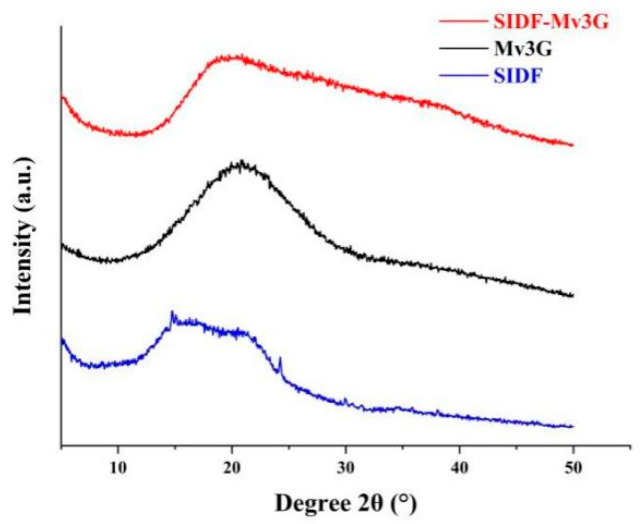
X-ray diffraction patterns. SIDF-Mv3G: the stability system of soybean insoluble dietary fiber and malvidin-3-*O*-glucoside; Mv3G: malvidin-3-*O*-glucoside; SIDF: soybean insoluble dietary fiber.

**Figure 7 foods-11-01474-f007:**
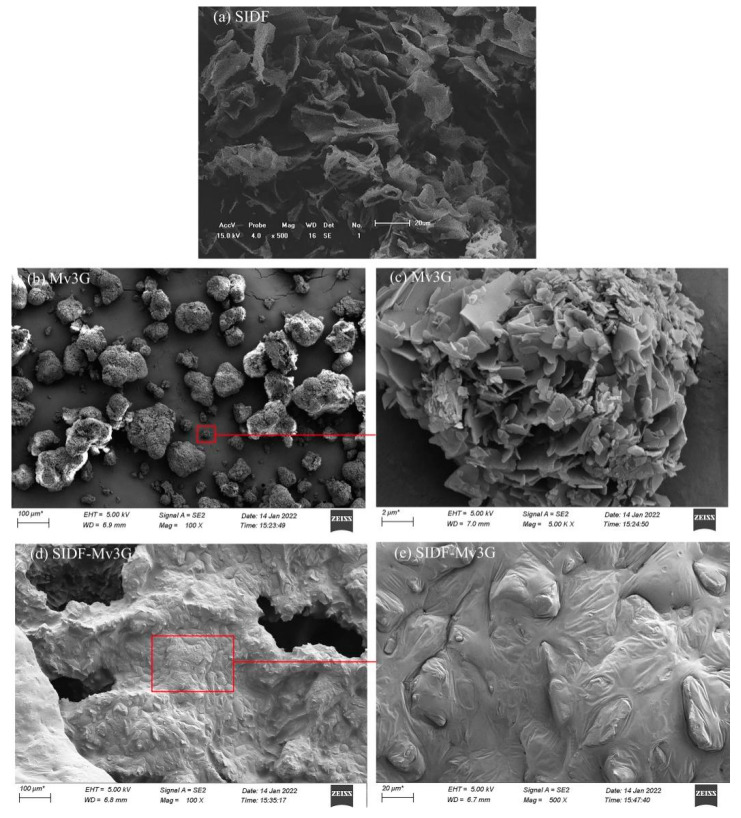
Scanning electron microscopy. SIDF-Mv3G: the stability system of soybean insoluble dietary fiber and malvidin-3-*O*-glucoside; Mv3G: malvidin-3-*O*-glucoside; SIDF: soybean insoluble dietary fiber. (**a**) Scanning electron microscope image of SIDF (×500); (**b**) scanning electron microscope image of Mv3G (×100); (**c**) scanning electron microscope image of Mv3G (×5000); (**d**) scanning electron microscope image of SIDF-Mv3G (×100); (**e**) scanning electron microscope image of SIDF-Mv3G (×500).

**Table 1 foods-11-01474-t001:** Results of orthogonal experiment.

Samples	A: SIDF ^1^–Mv3G ^2^ Ratio	B: SIDF Particle Size (Mesh Sieve)	C: Emulsification Temperature (°C)	D: Emulsification Time (min)	Absorbance	Precipitation Eccentricity (%)	Composite Scores
1	1 (1:20)	1 (200)	1 (20)	1 (30)	0.807	0.855	−1.128
2	1	2 (300)	2 (30)	2 (40)	0.880	0.786	1.2
3	1	3 (400)	3 (40)	3 (50)	0.855	0.800	0.195
4	2 (1:30)	1	2	3	0.885	0.781	1.246
5	2	2	3	1	0.850	0.786	0.719
6	2	3	1	2	0.830	0.857	−0.793
7	3 (1:40)	1	3	2	0.819	0.889	−1.507
8	3	2	1	3	0.828	0.829	−0.355
9	3	3	2	1	0.864	0.818	0.406
K_1_	0.27	−1.39	−2.28	0.00			
K_2_	1.17	1.56	2.85	−1.10			
K_3_	−1.46	−0.19	−0.59	1.09			
R	0.88	0.98	1.71	0.73			
Index		Average value		Standard deviation	Variable coefficient		Weight
Absorbance		0.846		0.026	0.030		0.411
Precipitation eccentricity		0.825		0.035	0.043		0.589
Samples	Standardized index						
	Absorbance			Precipitation eccentricity			
1	−1.520			−0.855			
2	1.325			1.112			
3	0.351			0.086			
4	1.520			1.054			
5	0.156			1.112			
6	−0.623			−0.912			
7	−1.052			−1.825			
8	−0.701			−0.114			
9	0.701			0.200			
Projects		Sums of squared deviations	Freedom	Mean square	F value		Significance
SIDF–Mv3G ratio		1.19	2	0.59	4.48		*
SIDF particle size		1.47	2	0.74	5.54		*
Emulsification temperature		4.56	2	2.28	17.16		**
Emulsification time		0.80	2	0.40	3.00		
Error		0.80	6	0.13			

SIDF ^1^: soybean insoluble dietary fiber; Mv3G ^2^: malvidin-3-*O*-glucoside; *: the difference was significant (*p* < 0.05); **: the difference was significant (*p* < 0.01).

**Table 2 foods-11-01474-t002:** Color measurement analysis.

	L*	a*	b*	∆E
pH 1 Mv3G solution	43.02 ^b^	9.34 ^a^	2.11 ^bc^	38.88 ^gh^
pH 1 Mv3G-SIDF ^1^	49.06 ^a^	10.14 ^a^	3.37 ^a^	38.31 ^h^
pH 2 Mv3G solution	43.18 ^b^	7.62 ^b^	1.43 ^c^	37.51 ^hi^
pH 2 Mv3G-SIDF	46.28 ^a^	9.15 ^a^	2.63 ^ab^	37.81 ^hi^
pH 3 Mv3G solution	42.88 ^b^	6.02 ^d^	−2.48 ^d^	37.69 ^hi^
pH 3 Mv3G-SIDF	43.84 ^b^	7.21 ^bc^	2.56 ^abc^	36.57 ^i^
pH 4 Mv3G solution	37.94 ^c^	5.73 ^d^	−7.20 ^f^	41.26 ^ef^
pH 4 Mv3G-SIDF	41.26 ^b^	7.38 ^bc^	−3.99 ^e^	39.95 ^fg^
pH 5 Mv3G solution	33.76 ^de^	3.96 ^e^	−8.43 ^g^	42.15 ^e^
pH 5 Mv3G-SIDF	35.45 ^cd^	6.61 ^cd^	−6.85 ^f^	42.62 ^de^
pH 6 Mv3G solution	31.69 ^ef^	2.79 ^f^	−12.32 ^h^	44.49 ^c^
pH 6 Mv3G-SIDF	32.76 ^ef^	6.07 ^d^	−7.81 ^fg^	43.80 ^cd^
pH 7 Mv3G solution	26.70 ^g^	2.64 ^f^	−14.83 ^i^	48.36 ^a^
pH 7 Mv3G-SIDF	30.59 ^f^	4.18 ^e^	−12.91 ^h^	46.27 ^b^

SIDF-Mv3G ^1^: the stability system solution of soybean insoluble dietary fiber and malvidin-3-*O*-glucoside. Continuous different letters indicate significant difference (*p* < 0.05).

## Data Availability

Data are contained within the article.

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
