# Peer review of "Protective Effect and Mechanism of Soybean Insoluble Dietary Fiber on the Color Stability of Malvidin-3-*O*-glucoside"

_foods, 2022, doi:10.3390/foods11101474_

Round 1

Reviewer 1 Report

[Foods] Manuscript ID: foods-1719743

 Protective effect and mechanism of soybean insoluble dietary fiber on the color stability of malvidin-3-O-glucoside

Reviewer Report:

The manuscript deals with the interaction between soy protein insoluble fiber and malvidin 3-glucosides and other anthocyanins present in Vitis amurensis Rupr fruits from China.

The concept is valid and there is an increase in the number of publications studding the fiber/polysaccharide and protein interactions with polyphenols and how this interaction affect the produced complex in terms of its stability and related human health properties. The methodology and tools to access the mechanisms of this interaction and stability of the produced complex are good and up to date.

Comments

Specific comment: The language and grammar need to revised because it affects the manuscript content quality and easy understanding of the meaning.

Title: Good title representing the content of the manuscript

Abstract:

The abstract represent a concise summary of the results obtained from this study. The information is vague, like different pH, high temperatures, sun light exposure, and whaich parameters gave you optimal conditions. Please provide details like range tested for pH and temperature, conditions related to sunlight exposure. Explain how UV and FTIR revealed the H-bonding formation.

Keywords

Soybean insoluble dietary fiber is present in the title, you should choose another keyword.

  1. Introduction

Line 39 needs citation

Lines 39-43: Long sentence, please rephrase to clear the meaning.

Line 50, “which can improved the stability of polyphenols” review the grammar

Line 55, please explain what is an inclusion complex

Lines 69-72, review for grammar and proper meaning of the sentence

Line 77-78, application of anthocyanin as 77 natural pigment or functional component (in what?)

2- Materials & Methods

Line 89 , All other chemical fractions used are of analytical grade.(what do you mean by chemical fractions)

Line 140, date of (what is it?)

  • Results

Line 164-166, please review the sentence for clear meaning

Line 201, the taste of food to decrease (Change to taste quality of food decreased)

Line 211, flavonoids, replace with anthocyanins

Author Response

Response to Reviewer 1 Comments

Point 1: The abstract represent a concise summary of the results obtained from this study. The information is vague, like different pH, high temperatures, sun light exposure, and whaich parameters gave you optimal conditions. Please provide details like range tested for pH and temperature, conditions related to sunlight exposure. Explain how UV and FTIR revealed the H-bonding formation.

Response 1: Thanks for your correction. We revised the contents of the abstract. “The optimum condition of SIDF and Mv3G stable system at pH 3.0 was determined by orthogonal experiment. The results indicated that SIDF effectively improved the stability of Mv3G under different pH, high temperature and sunlight conditions. UV-Vis spectrophotometry and Fourier transform infrared spectroscopy analyses revealed that the interaction between SIDF and Mv3G was due to hydrogen bonding.” Has been changed to “The optimum condition of SIDF and Mv3G stable system at pH 3.0 was determined by orthogonal experiment. The results indicated that SIDF effectively improved the stability of Mv3G under different pH (1.0~7.0), high temperature (100℃ for 100 min) and sunlight (20 ± 2℃ for 30 d) conditions. The absorption peak intensity of the UV-Vis spectrum of SIDF-Mv3G was enhanced. Fourier transform infrared spectroscopy analyses revealed that the -OH stretching vibration peak of SIDF-Mv3G was changed, which indicated that the interaction between SIDF and Mv3G was due to hydrogen bonding.” (Line 18-25)

Point 2: Soybean insoluble dietary fiber is present in the title, you should choose another keyword. 

Response 2: “Anthocyanins; Soybean insoluble dietary fiber; Inclusion complex; Hydrogen bonding” has been changed to “Anthocyanins; Insoluble dietary fiber; Stable system; Inclusion complex; Hydrogen bonding” (Line 31-32)

Point 3: Line 39 needs citation.

Response 3: “The distribution of the six most common anthocyanins in nature is cyandin, delphinidin, pelargonidin, peonidin, petunidin and malvidin, which accounting for 50%, 12%, 12%, 12%, 7%, 7% of the total anthocyanins, respectively [5].” There were references in line 39. (Line 42)

Point 4: Lines 39-43: Long sentence, please rephrase to clear the meaning.

Response 4: “The distribution of the six most common anthocyanins in nature is cyandin, delphinidin, pelargonidin, peonidin, petunidin and malvidin, which accounting for 50%, 12%, 12%, 12%, 7%, 7% of the total anthocyanins, respectively [5]. Malvidin has high biological activity, but the overall distribution is less. It is worth noting that the high-yield Vitis amurensis Rupr in the Changbai Mountains of China are rich in anthocyanins (the average content is above 150 mg/100g·FW, and individual varieties are as high as 400 mg/100g·FW), among of the content of malvidin is rich accounting for 55~65% of total anthocyanins, and malvidin-3-O-glucoside (Mv3G) monomer is the most [6].” has been changed to “The distribution of the six most common anthocyanins in nature is cyandin, delphinidin, pelargonidin, peonidin, petunidin and malvidin, which accounting for 50%, 12%, 12%, 12%, 7%, 7% of the total anthocyanins, respectively [5]. Malvidin has high biological activity, but the overall distribution is less in nature [5]. It is worth noting that the high-yield Vitis amurensis Rupr in the Changbai Mountains of China are rich in anthocyanins (the average content is above 150 mg/100g·FW, and individual varieties are as high as 400 mg/100g·FW) [6,7]. Malvidin accounts for 55~65% of total anthocyanins from Vitis amurensis Rupr, among which malvidin-3-O-glucoside (Mv3G) is the most abundant monomer [8]” (Line 40-48)

Point 5: Line 50, “which can improved the stability of polyphenols” review the grammar.

Response 5: “During the food processing, polyphenols can interact with starch, protein and cellulose to form a complex, which can improved the stability of polyphenols” has been changed to “During the food processing, polyphenols can interact with starch, protein and cellulose to form a complex, which can improve the stability of polyphenols ” (Line 53-54)

Point 6: Line 55, please explain what is an inclusion complex.

Response 6: The inclusion complex (A mixture in which the molecules of one component are contained in the crystal lattice of another component.). (Line 59-61)

Point 7: Lines 69-72, review for grammar and proper meaning of the sentence.

Response 7: “SIDF has loose structure and rough surface, which can promote the interaction with polyphenols by adsorption or embedding [20]. Zhao et al. [21] had been reported that smaller particle size significantly increased the water-holding capacity of insoluble dietary fiber (IDF), because micronization reduced the interfacial tension and could exposes more polar groups, surface area and water-binding sites to the surrounding water.” has been changed to “SIDF has loose structure and rough surface, which can promote the interaction with polyphenols by adsorption or embedding [20]. Zhao et al. [21] had been reported that the hydration of insoluble dietary fiber (IDF) improved as its particle size decreased because of the greater surface area, increased number of polar groups and the exposure of other waterbinding sites of IDF to the surrounding water.” (Line 73-78)

Point 8: Line 77-78, application of anthocyanin as 77 natural pigment or functional component (in what?).

Response 8: “In order to further explore the interaction between SIDF and Mv3G, this study explained the stabilization mechanism of SIDF and Mv3G, which can provide the possibility for the application of anthocyanin as natural pigment or functional component.” has been changed to “In order to further explore the interaction between SIDF and Mv3G, this study explained the stabilization mechanism of SIDF and Mv3G, which will be more conducive to anthocyanins as stable pigments or functional components added to food.” (Line 81-85)

Point 9: Line 89 , All other chemical fractions used are of analytical grade.(what do you mean by chemical fractions).

Response 9: “All other chemical fractions used are of analytical grade.” has been changed to “All chemicals used in this work were all analytical or HPLC grade.” (Line 95-96)

Point 10: Line 140, date of (what is it?)

Response 10: “XRD date of SIDF-Mv3G, SIDF and Mv3G were collected using an X-ray diffractometer (D8 ADVANCE, Germany), equipped with copper Cu Kα radiation (λ = 0.154 nm) as the X-ray source (2θ range of 5~50°).” has been changed to “XRD diffraction pattern of SIDF-Mv3G, SIDF and Mv3G were collected using an X-ray diffractometer (D8 ADVANCE, Germany), equipped with copper Cu Kα radiation (λ = 0.154 nm) as the X-ray source (2θ range of 5~50°).” (Line 152-154)

Point 11: Line 164-166, please review the sentence for clear meaning 

Response 11: “The variance analysis showed that emulsification temperature was very significantly, SIDF particle size and SIDF:Mv3G ratio exhibited was significantly, emulsification time was not significantly on the stable system.” has been changed to “The variance analysis showed that the effects of the single factors of emulsification temperature were significant (P < 0.01) on the stable system, and SIDF particle size and SIDF:Mv3G ratio exhibited were significant on the stable system (P < 0.05), whereas emulsification time was not significant on the stable system.” (Line 176-179)

Point 12: Line 201, the taste of food to decrease (Change to taste quality of food decreased).

Response 12: “At the same time, the lower pH value could cause the taste of food to decrease.” has been chenged to “At the same time, the lower pH value could cause taste quality of food decreased.” (Line 215-216)

Point 13: Line 211, flavonoids, replace with anthocyanins.

Response 13: “Anthocyanins degraded rapidly at high temperature, and the structure of flavonoids was opened and converted into chalcone, resulting in unpleasant brown substances [28]” has been changed to “Anthocyanins degraded rapidly at high temperature, and the structure of anthocyanins was opened and converted into chalcone, resulting in unpleasant brown substances [28]” (Line 225-226)

Reviewer 2 Report

In this manuscript authors have looked at the protective effect of soybean insoluble fiber on malvidin-3-o-glucoside. A few problem with this manuscript that require proper mention

1. Since authors used insoluble fiber for the experiment, turbidity problems are expected to occur when measured '2.3.2 thermal stability' and '2.3.3.sunlight stability' using a UV-Vis spectrophotometer.  A detailed explanation of the methods of '2.3.2' & '2.3.3' are needed.

2. Line 94 : the method for emulsification should be described.

3. Line 175 :  Remove "mean ± SD" since the numbers written with SD are not shown in Table 1.

Author Response

Response to Reviewer 2 Comments

Point 1: Since authors used insoluble fiber for the experiment, turbidity problems are expected to occur when measured '2.3.2 thermal stability' and '2.3.3.sunlight stability' using a UV-Vis spectrophotometer. A detailed explanation of the methods of '2.3.2' & '2.3.3' are needed.

Response 1: Thanks for your correction. We added a detailed test method. Because dietary fiber is not added much, it will not be too turbid. “The samples were prepared in 5 mL glass tubes wrapped in tin foil, which were heated in a water bath at 100°C for 100 min. Every 20 min, the samples were quickly cooled in ice. After 5 min,the absorbance of upper supernatant was measured at 521 nm by UV-Vis spectrophotometry (the proper amount of dietary fiber will not be too turbid). The retention rate (R, %) of anthocyanins was used as the evaluation index to evaluate the thermal stability.” (Line 126-131)

Seal SIDF-Mv3G and Mv3G solution in vials, respectively, and sunlight illumination at room temperature (20 ± 2°C, samples were placed on the indoor windowsill and the daily insolation time was not less than 6 h). Samples were taken out after 0, 5, 10, 15, 20, 25 and 30 d, respectively. The absorbance of upper supernatant was measured at 521 nm by UV-Vis spectrophotometry (the proper amount of dietary fiber will not be too turbid) and the retention rate of anthocyanins was calculated according to Eq. (3) (Line 135-140)

Point 2: Line 94 : the method for emulsification should be described.

Response 2: We added an experimental process. “Weigh the right amount of SIDF and Mv3G which were mixed in a certain proportion with PBS (citric acid-disodium hydrogen phosphate) at pH 3.0 to 100 mL (the final Mv3G concentration was 0.12 mg/100mL). Under a certain temperature range, the stabilized system was prepared by stirring and emulsifying at 500 rmp/min..” (Line 98-101)

Point 3: Line 175 : Remove "mean ± SD" since the numbers written with SD are not shown in Table 1.

Response 3: “This result had been confirmed by three parallel validations, and the actual score for the optimal condition was 1.387 ± 0.034” has been changed to “This result had been confirmed by three parallel validations, and the actual score for the optimal condition was 1.387” (Line 180-181)

Round 2

Reviewer 1 Report

The authors have done good job in responding to all reviewer's comments, and language has been improved.